# Dynamic inference of cell developmental complex energy landscape from time series single-cell transcriptomic data

**Qi Jiang**[1,2], **Shuo Zhang**[1,2], **Lin Wan**[1,2]*

**1** NCMIS, LSC, LSEC, Academy of Mathematics and Systems Science, Chinese Academy of Sciences, Beijing, China, **2** School of Mathematical Sciences, University of Chinese Academy of Sciences, Beijing, China

* lwan@amss.ac.cn

**Data Availability Statement:** GraphFP software is available at https://github.com/QiJiang-QJ/GraphFP.

## Abstract

Time series single-cell RNA sequencing (scRNA-seq) data are emerging. However, dynamic inference of an evolving cell population from time series scRNA-seq data is challenging owing to the stochasticity and nonlinearity of the underlying biological processes. This calls for the development of mathematical models and methods capable of reconstructing cellular dynamic transition processes and uncovering the nonlinear cell-cell interactions. In this study, we present GraphFP, a nonlinear Fokker-Planck equation on graph based model and dynamic inference framework, with the aim of reconstructing the cell state-transition complex potential energy landscape from time series single-cell transcriptomic data. The free energy of our model explicitly takes into account of the cell-cell interactions in a nonlinear quadratic term. We then recast the model inference problem in the form of a dynamic optimal transport framework and solve it efficiently with the adjoint method of optimal control. We evaluated GraphFP on the time series scRNA-seq data set of embryonic murine cerebral cortex development. We illustrated that it 1) reconstructs cell state potential energy, which is a measure of cellular differentiation potency, 2) faithfully charts the probability flows between paired cell states over the dynamic processes of cell differentiation, and 3) accurately quantifies the stochastic dynamics of cell type frequencies on probability simplex in continuous time. We also illustrated that GraphFP is robust in terms of cluster labelling with different resolutions, as well as parameter choices. Meanwhile, GraphFP provides a model-based approach to delineate the cell-cell interactions that drive cell differentiation. GraphFP software is available at https://github.com/QiJiang-QJ/GraphFP.

## Author summary

Dynamic inference of cell development processes from time series scRNA-seq data is a major challenge. Here, we present GraphFP, a coherent computational framework that simultaneously reconstructs the cell state-transition complex potential energy landscape and infers cell-cell interactions from time series single-cell transcriptomic data. Based on the mathematical framework of nonlinear Fokker-Planck equation on graph, GraphFP

**Funding:** This work was supported by the Fund to LW from the National Key Research and Development Program of China under Grant 2019YFA0709501. LW and SZ were also supported by the National Natural Science Foundation of China (No. 12071466 to LW and No.11871465 to SZ). The funders had no role in study design, data collection and analysis, decision to publish, or preparation of the manuscript.

**Competing interests:** The authors have declared that no competing interests exist.

models the stochastic dynamics of the cell state/type frequencies on probability simplex in continuous time, where the free energy with a nonlinear quadratic interaction term is employed to characterize cell-cell interactions. We formulate the model inference problem in the form of a dynamic optimal transport framework and solve it efficiently with the celebrated adjoint method. GraphFP allows for 1) reconstructing cell state potential energy, which is a measure of cellular differentiation potency, 2) charting the probability flows between paired cell states over dynamic processes, 3) quantifying the stochastic dynamics of cell type frequencies on probability simplex in continuous time, and 4) delineating cell-cell interactions that drive cell differentiation. We show how GraphFP can be used to faithfully reveal and accurately quantify the cell development processes using the embryonic murine cerebral cortex development time series scRNA-seq dataset.

## Introduction

The dynamics of cell developmental processes (e.g., cell differentiation and tumorigenesis) are highly complex. Advances in single-cell RNA sequencing (scRNA-seq) technologies enable cell-resolved investigation of heterogeneous cell populations, offering a systematic approach to reveal underlying developmental dynamics, cell communication, and gene regulation [1]. The dynamic inferences of cell development from scRNA-seq transcriptomic profiles draw heavily on advances in computational and systems biology. Many efforts have been advanced to reconstruct cell developmental trajectories and pseudo-time from the single-cell snapshot profile sampled from an evolving cell population [2]. Methods have also been developed to quantify cell developmental landscape [3–6]. However, state-of-the-art dynamic inference methods based on the static single-cell snapshot profile alone may lack identifiability of complex dynamic processes [7].

Recently, time series scRNA-seq data profiled from cells sampled at multiple physical time stages have been accumulating, accounting for additional temporal dimension. The wider dynamic ranges enriched by the temporal dimension show great promise in overcoming the difficulties that arise during the inferences from static single-cell snapshot profiling. Computational methods that explicitly incorporate temporal information have been developed. TASIC determined the temporal trajectories based on the probabilistic graphical model to integrate expression and time information [8], while CSHMM developed a continuous state hidden Markov model to infer trajectory structure and assigned cells to paths [9]. TSEE incorporated temporal information into a nonlinear dimensionality reduction algorithm of elastic embedding to visualize dynamic gene expression patterns, offering enhanced temporal resolution [10]. ScPADGRN reconstructed the dynamic gene regulatory network with a preconditioned ADMM optimization method [11]. Tempora incorporated biological pathway information to accurately identify cell types and then incorporated the time information to infer evolving cell-type trajectories [12].

An emerging number of methods are being developed to reconstruct cell developmental energy landscape from time series single-cell data using the mathematical framework of optimal transport. Optimal transport has received considerable attention in recent years for various disciplines such as machine learning and statistical data analysis, as it has been proven to be a powerful tool in the analysis of complex data [13]. The core concept of optimal transport, Wasserstein distance between two probability distributions, quantifies an optimal cost of transporting one data distribution to the other. As a remarkably rich and fruitful concept, Wasserstein distance "enables a mechanism transforming the probability space into a

Riemannian manifold (known as a Wasserstein manifold), so that geometric structures and partial differential equation (PDE) techniques can be established and analyzed" [14]. Amongst existing methods, Waddington-OT [15] reported landmark work that developed an unbalanced optimal transport framework to reconstruct the cell developmental landscape by inferring cell-cell probabilistic couplings based on the distributions between adjacent time points [15]; TrajectoryNet set up a dynamic optimal transport neural network framework to reconstruct the continuous normalizing flows of evolving cell populations on the continuous state space [16]; PRESCIENT modelled cell differentiation as a diffusion process over a potential energy landscape learned by the neural network framework [17]. However, the computation of optimal transport is still a bottleneck when processing large-scale data [13].

In this study, we present GraphFP, a nonlinear Fokker-Planck equation on graph based model and dynamic inference framework, with the aim of reconstructing the cell state-transition complex potential energy landscape from time series single-cell transcriptomic data. The Fokker-Planck equation (FPE) is ubiquitously applied in the modelling of statistical physics and biological systems, including single-cell data analysis [5, 7, 18]. GraphFP is built on the mathematical framework established by the FPE on finite graph originally introduced by Chow *et al.* [19, 20] and Li [21] (see [14] for a recent survey). Building upon the fundamental form of optimal transport, GraphFP learns the complex geometry of data, as well as provides a novel way to quantify cell-cell interactions during cell development. It models the cell developmental process as stochastic dynamics of the cell state/type frequencies on probability simplex in continuous time. The discrete Wasserstein distance is introduced to transform the probability simplex into a Riemannian manifold, called discrete Wasserstein manifold. The FPE is proven to be the gradient flow of the free energy on the discrete Wasserstein manifold. The free energy of our model consists of a static linear potential energy term and a nonlinear quadratic interaction energy term that characterizes cell-cell interactions [22, 23]. To estimate the parameters which represent the linear potential energy and cell-cell interaction strengths, we recast the model inference problem in the form of a dynamic optimal transport framework and solve it efficiently with the celebrated adjoint method of optimal control [24]. The adjoint method also played a central role in the development of the well-known neural network algorithm NeuralODE [25].

We evaluated GraphFP on the time series scRNA-seq dataset of embryonic murine cerebral cortex development [26]. GraphFP reconstructed the cell state potential energy, which is a measure of cellular differentiation potency from both static and dynamic points of view. It faithfully charted the probability flows of cell state-transitions, consistent with the gold standard benchmarks. It also accurately quantified the stochastic dynamics of cell type frequencies on probability simplex in continuous time. GraphFP delineated cell-cell interactions that drive cell development in a model-based fashion. We tested the cell-cell interaction term of GraphFP by illustrating its ability to fit the nonlinear curves of experimental data and recover held-out time points. We illustrated that GraphFP is robust in terms of cluster labelling with different resolutions, as well as parameter choices. We compared GraphFP with state-of-the-art cell developmental energy landscape reconstruction methods in **Discussion**.

## Methods

GraphFP is a coherent computational framework that simultaneously reconstructs the cell state-transition complex potential energy landscape and infers cell-cell interactions from time series single-cell transcriptomic data (Fig 1). It models cell state-transition dynamics with cell-cell interactions based on the mathematical framework introduced by Chow *et al.* [19, 20] and Li [21].

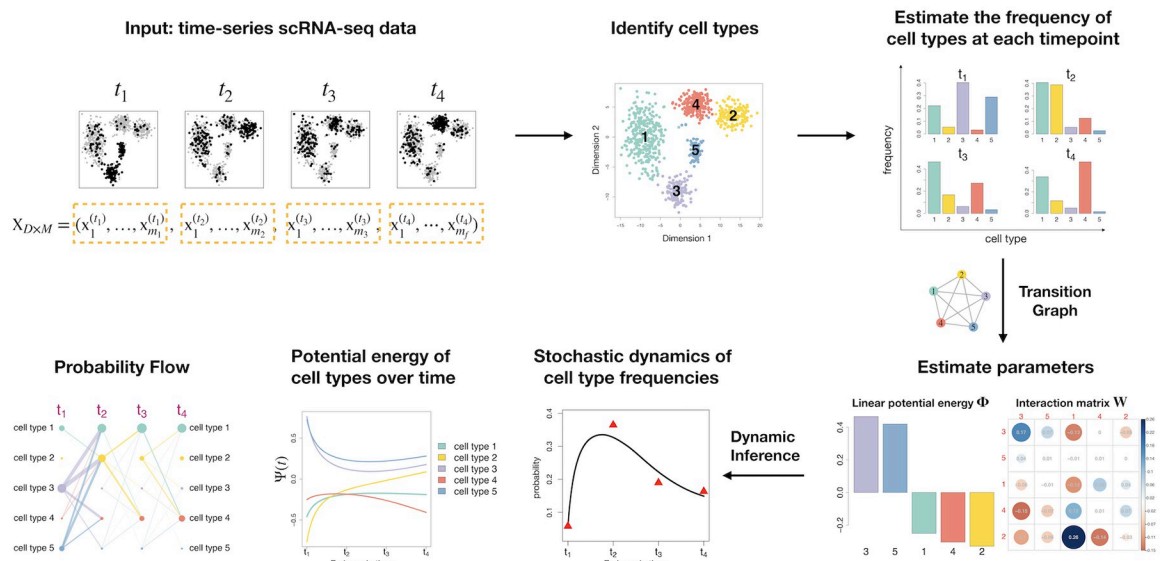

**Fig 1. Overview of GraphFP algorithm.** GraphFP takes the input of time series single-cell transcriptomic data incorporated with experimental temporal information. It identifies cell states/types, estimates the cell type frequencies at each time point, estimates the linear potential energy $\mathbf{\Phi}$ and the cell-cell interaction matrix $W$ based on the adjoint method. GraphFP outputs the stochastic dynamics of cell type frequencies $\boldsymbol{p}(t)$ on probability simplex in continuous time, the cell state transition potential energy, and the probability flows of cell state-transitions underlying the evolving cell population.

GraphFP allows for reconstructing cell state potential energy, charting the probability flows between paired cell states over dynamic process, quantifying the stochastic dynamics of cell type frequencies on probability simplex in continuous time, and delineating cell-cell interactions that drive cell differentiation [22]. The main steps for GraphFP are organized in the following subsections.

## Identifying cell states/types

Given the time series scRNA-seq data, the single-cell samples are collected at $f$ time stages $\{t_1, t_2, \ldots, t_f\}$, and for each time stage $t_l(1 \le l \le f)$, a number of $m_l$ single cells are sequenced with the corresponding gene expression vectors $\boldsymbol{x}_1^{(t_l)}, \ldots, \boldsymbol{x}_{m_l}^{(t_l)} \in \mathbb{R}^D$, where $D$ is the number of genes for single cells. Thus, the single cell gene expression profile of a total number of $M = \sum_{l=1}^{f} m_l$ cells is contained in the data matrix

$$\boldsymbol{X}_{D \times M} = \left(\boldsymbol{x}_1^{(t_1)}, \ldots, \boldsymbol{x}_{m_1}^{(t_1)}, \ldots, \ldots, \boldsymbol{x}_1^{(t_f)}, \cdots, \boldsymbol{x}_{m_f}^{(t_f)}\right).$$

Suppose that the total number of $M$ cells forms $n$ cell states corresponding to $n$ cell types. When the cells are already annotated or clustered, GraphFP directly utilizes the prior information of cell types. When cell type information is not available, GraphFP will apply state-of-the-art single-cell clustering and annotation methods (e.g., Louvain-Jaccard algorithm [27], the single-cell data analysis platform Seurat [28], and the single cell deep generative method scDEC [29]) to cluster cells into $n$ clusters as the cell states/types.

## Constructing the cell state-transition graph

The cell state-transition graph $G = (V, E)$ is an undirected graph, where each vertex $i$ in $V$ represents a cell state/type and each edge $\{i, j\}$ in $E$ represents that cell state $i$ can directly transit to

cell state $j$ or vice versa. In this study, with no inherent reliance on any prior information, we construct the cell state-transition graph $G$ as a complete graph supported on all cell types. However, we can also incorporate prior information of cell state-transition into graph $G$ when available.

## Modelling cell state-transition dynamics with the nonlinear FPE on graph equipped with discrete $L_2$-Wasserstein distance

GraphFP models the stochastic dynamics of cell state/type frequencies in the evolving cell population with the nonlinear FPE on graph. The underlying assumption of this model is that cell state-transitions follow the minimum total kinetic energy path during cell differentiation, which can be measured by the discrete $L_2$-Wasserstein distance on graph. To estimate the parameters in the nonlinear FPE of GraphFP, we propose a gradient method to find the parameters that minimize the discrete $L_2$-Wasserstein distance.

We use discrete probability distributions supported on the vertices of cell state-transition graph $G$ to represent the state of the system at time $t$. Suppose the number of vertices of $G$ is $n$, the set of system states is the probability simplex supported on all vertices of $G$:

$$\mathcal{P}(G) = \left\{ \boldsymbol{p} = \boldsymbol{p}(t) = (p_i(t))_{i=1}^n \left| \sum_{i=1}^n p_i(t) = 1, p_i(t) \geq 0 \right. \right\}.$$

The cell-state frequencies or probabilities are estimated for each time point separately, resulting in $\hat{\boldsymbol{p}}(t_1) = \boldsymbol{p}^1, \hat{\boldsymbol{p}}(t_2) = \boldsymbol{p}^2, \cdots, \hat{\boldsymbol{p}}(t_f) = \boldsymbol{p}^f$.

Here, we mainly follow the setups in Chow *et al.* [20]; Li [21], and define the discrete nonlinear free energy $\mathcal{F} : \mathcal{P}(G) \to \mathbb{R}$ of the cell-state system as follows:

$$
\begin{aligned}
\mathcal{F}(\boldsymbol{p}|\boldsymbol{\Phi}, \boldsymbol{W}) &= \mathcal{V}(\boldsymbol{p}) + \mathcal{W}(\boldsymbol{p}) + \beta\mathcal{H}(\boldsymbol{p}), \\
&= \sum_{i=1}^n \Phi_i p_i + \frac{1}{2}\sum_{i=1}^n\sum_{j=1}^n w_{ij} p_i p_j + \beta\sum_{i=1}^n p_i \log p_i, \\
&= \boldsymbol{p}^T\boldsymbol{\Phi} + \frac{1}{2}\boldsymbol{p}^T\boldsymbol{W}\boldsymbol{p} + \beta\sum_{i=1}^n p_i \log p_i,
\end{aligned}
\tag{1}
$$

where $\mathcal{V}$, $\mathcal{W}$, and $\mathcal{H}$ represent the static linear potential energy parametrized by vector $\boldsymbol{\Phi} = (\Phi_i)_{i=1}^n$, the quadratic interaction energy of paired cell states parametrized by matrix $\boldsymbol{W} = (w_{ij})_{1 \leq i, j \leq n}$, and Boltzmann entropy with a hyper-parameter $\beta \geq 0$, respectively. In general, the interaction matrix $\boldsymbol{W}$ is asymmetry, with its element $w_{ij}$ representing the interaction strength from cell state $j$ (as a signalling sender) to cell state $i$ (as a signalling receiver) (see Eq (9) and section "**Reconstruction of cell developmental energy landscape and modelling of cell-cell interactions**" for further discussion). We hereinafter denote the parameters of the free energy as $\boldsymbol{\theta} \equiv \{\boldsymbol{\Phi}, \boldsymbol{W}\}$.

Based on the specific form of free energy $\mathcal{F}$, the corresponding FPE on $G$ is defined as follows: for any $i \in V$,

$$\frac{dp_i(t)}{dt} = \sum_{j \in N(i)} (F_j(\boldsymbol{p}(t)) - F_i(\boldsymbol{p}(t)))g_{ij}(\boldsymbol{p}(t)), \tag{2}$$

where $F_i(\boldsymbol{p}(t)) \equiv \frac{\partial \mathcal{F}(\boldsymbol{p}(t))}{\partial p_i}$, $N(i) = \{j \in V | \{i, j\} \in E\}$ is the adjacency set of vertex $i \in V$, and

$g_{ij}(\boldsymbol{p}(t))$ is defined as

$$
g_{ij}(\boldsymbol{p}(t)) = \begin{cases} p_j(t) & \text{if } F_j(\boldsymbol{p}(t)) > F_i(\boldsymbol{p}(t)), j \in N(i); \\ p_i(t) & \text{if } F_j(\boldsymbol{p}(t)) < F_i(\boldsymbol{p}(t)), j \in N(i); \\ \dfrac{p_j(t) + p_i(t)}{2} & \text{if } F_j(\boldsymbol{p}(t)) = F_i(\boldsymbol{p}(t)), j \in N(i). \end{cases} \tag{3}
$$

Chow *et al.* [20] and Li [21] proved that, the dynamics of $\boldsymbol{p}(t)$ is evolving along the gradient of free energy (Eq (1)) when $\mathcal{P}(G)$ is equipped with the discrete $L_2$-Wasserstein distance on graph $G$: for any $\boldsymbol{p}^1, \boldsymbol{p}^f \in \mathcal{P}(G)$,

$$
\mathcal{W}^2_{2;\mathcal{F}}(\boldsymbol{p}^1, \boldsymbol{p}^f) = \inf \int_{t_1}^{t_f} \frac{1}{2} \sum_{\{i,j\} \in E} (F_j(\boldsymbol{p}(t)) - F_i(\boldsymbol{p}(t)))^2 \cdot g_{ij}(\boldsymbol{p}(t)) dt,
$$

where the infimum is taken over $\mathcal{C} = \{\boldsymbol{p}(t) \in \mathcal{P}(G)$ is a piecewise $C^1$ curve that satisfies the FPE of Eq (2), $\boldsymbol{p}(t_1) = \boldsymbol{p}^1$ and $\boldsymbol{p}(t_f) = \boldsymbol{p}^f\}$. Intuitively, the discrete $L_2$-Wasserstein distance can be understood as the total work for transporting $\boldsymbol{p}^1$ to $\boldsymbol{p}^f$ on $\mathcal{P}(G)$, which is the summation of the kinetic energies of the flows (mass×squared velocity) on the edges of graph $G$ during the time period $[t_1, t_f]$.

## Parameter estimation and model optimization

To estimate the parameters $\boldsymbol{\theta} = \{\boldsymbol{\Phi}, W\}$ of the free energy using all time series scRNA-seq data collected at time points $\{t_1, t_2, \cdots, t_f\}$ (assume that $\boldsymbol{\theta}$ is constant over the entire time period), we formulate the estimation problem as a minimization problem of the discrete $L_2$-Wasserstein distance:

$$
\boldsymbol{\theta}^* = \arg\min_{\boldsymbol{\theta}} \quad \int_{t_1}^{t_f} \frac{1}{2} \sum_{\{i,j\} \in E} (F_i(\boldsymbol{p}(t)) - F_j(\boldsymbol{p}(t)))^2 \cdot g_{ij}(\boldsymbol{p}(t)) dt,
$$

subject to the constraints

$$
\begin{aligned}
\frac{d\boldsymbol{p}(t)}{dt} &= \left( \sum_{j \in N(i)} (F_j(\boldsymbol{p}(t)) - F_i(\boldsymbol{p}(t))) g_{ij}(\boldsymbol{p}(t)) \right)_{i=1}^{n}, \\
\boldsymbol{p}(t_l) &= \boldsymbol{p}^l, l = 1, 2, \ldots, f,
\end{aligned} \tag{4}
$$

where $n$ is the size of vertex set $V$. However, the dynamic optimization problem with constrains of multiple fixed points at $\boldsymbol{p}(t_l)$s in Eq (4) is extremely difficult, and maybe even unsolvable.

In this study, to estimate parameters $\boldsymbol{\theta}^*$, we follow TrajectoryNet [16] and relax the constraints of $\boldsymbol{p}$s at time points $\{t_2, \cdots, t_f\}$ by moving them into the object function as follows

$$
\boldsymbol{\theta}^* = \arg\min_{\boldsymbol{\theta}} \left[ \int_{t_1}^{t_f} \frac{1}{2} \sum_{\{i,j\} \in E} (F_i(\boldsymbol{p}(t)) - F_j(\boldsymbol{p}(t)))^2 \cdot g_{ij}(\boldsymbol{p}(t)) dt + \sum_{l=2}^{f} \lambda_l \text{KL} \left( \boldsymbol{p}(t_l) \parallel \boldsymbol{p}^l \right) \right], \tag{5}
$$

subject to the constraints

$$\frac{d\boldsymbol{p}(t)}{dt} = \left(\sum_{j \in N(i)} (F_j(\boldsymbol{p}(t)) - F_i(\boldsymbol{p}(t)))g_{ij}(\boldsymbol{p}(t))\right)_{i=1}^n, \tag{6}$$

$$\boldsymbol{p}(t_1) = \boldsymbol{p}^1, \tag{7}$$

where $\lambda_l \geq 0$ is a constant regularization parameter, $n$ is the size of vertex set $V$, and

$\mathrm{KL}(\boldsymbol{p} \parallel \boldsymbol{q}) \equiv \sum_{i=1}^n p_i \log \frac{p_i}{q_i}$ is the Kullback-Leibler divergence between the probability distributions $\boldsymbol{p}$ and $\boldsymbol{q}$.

The optimization problem of Eq (5) can be interpreted as an optimal control problem with fixed initial point and the parameters $\boldsymbol{\theta}$ can be regarded as the control. We therefore propose a gradient method to estimate $\boldsymbol{\theta}^*$ based on the Pontryagin's Maximum Principle (also known as the adjoint method) [24] to solve the optimal control problem of Eq (5). In our implementation, we treat the integral part and the KL divergence part in Eq (5) separately, solve each of them using the adjoint method, and then combine them together through the tradeoff parameters $\lambda_l$s. See the details and the pseudocode of GraphFP algorithm in S1 Text.

In our implementation, we centralize both $\boldsymbol{\Phi}$ and $\boldsymbol{W}$ such that

$$\sum_{i=1}^n \Phi_i = 0, \quad \sum_{i,j=1}^n w_{ij} = 0.$$

Therefore, in our study, $w_{ij}$ with a high absolute value indicates strong interaction from cell state $j$ to cell state $i$ (see the following section for further discussion).

## Reconstruction of cell developmental energy landscape and modelling of cell-cell interactions

Following Chow *et al.* [20] and Li [21], we define the dynamic potential energy of cell type $i$ as follows:

$$\Psi_i(t) \equiv F_i(\boldsymbol{p}(t)) = \Phi_i + \sum_{k=1}^n w_{ik}p_k(t) + \beta \log p_i(t), \tag{8}$$

which consists of three components: (1) a static linear potential energy $\Phi_i$ that is time-independent and measures the cell differentiation potency of cell type $i$; (2) an interaction potential energy $\sum_{k=1}^n w_{ik}p_k(t)$ that coordinates cell development through intercellular communication; and (3) an entropy energy $\beta \log p_i(t)$ which is an analog of the potential energy induced by white noise in diffusion process [14].

Overall, the $\Psi_i(t)$ depicts a dynamic potential energy landscape of cell type $i$ at time $t$: cell state $i$ with a higher potential energy $\Psi_i(t)$ tends to transit to more stable states with lower potential energies; while the $\Phi_i$ quantifies the cell differentiation potency of cell type $i$, as well as a way to represent cell developmental time (see **The linear potential energy Φ by GraphFP quantifies cell differentiation potency** in **Results**).

Cell development relies on the coordination of cellular activities based on temporal and local cell-cell communication through molecular signalling events [22]. As a key component, the interaction potential energy $\sum_{k=1}^n w_{ik}p_k(t)$ in Eq (8) models and quantifies cell-cell interactions that drive cell development, where $w_{ik}$ is the interaction strength from cell type $k$ (as a signalling sender) to cell type $i$ (as a signalling receiver): when $w_{ik} > 0$, cell type $k$ will send

signals to upgrade the potential energy of cell type $i$ to a higher level, thus inhibiting the decrease of potential energy of cell type $i$; when $w_{ik} < 0$, cell type $k$ will send signals to downgrade the potential energy of cell type $i$ to a lower level, thus stimulating the decrease of potential energy of cell type $i$; when $w_{ik} = 0$ or $w_{ik} \approx 0$, cell type $k$ will send no or weak signals to cell type $i$, thus unable to alter the potential energy level of cell type $i$.

$$w_{ik} \begin{cases} > 0, & k \text{ sends signals to upgrade the potential of } i; \\ = 0/\approx 0, & k \text{ sends no or weak signals to } i, \text{ unable to alter its potential;} \\ < 0, & k \text{ sends signals to downgrade the potential of } i. \end{cases} \quad (9)$$

We say that cell types $i$ and $k$ have no mutual interactions when both $w_{ik}$ and $w_{ki}$ are zero or close to zero. Such modelling of the interaction matrix $W$ is inspired and evidenced by our increasing understanding that both positive and negative feedback circuits composed of stimulatory and inhibitory factors are involved in the regulation of precise coordination of cell fate decisions through intercellular communication [22, 30].

### Dynamic inference of cell developmental process

Once we estimate parameters $\theta^* = \{\Phi^*, W^*\}$, we can quantify the stochastic dynamics of the cell type frequencies $p(t)$ on probability simplex in continuous time $t(> t_1)$, according to Eq (2) given the initial point of probability $p^1$ on probability simplex.

The potential energy difference between cell states $i$ and $j$ is

$$\begin{aligned} \Delta_{ij}(t) &= \Psi_i(t) - \Psi_j(t) = F_i(p(t)) - F_j(p(t)) \\ &= (\Phi_i - \Phi_j) + \sum_{k=1}^{n} (w_{ik} - w_{jk})p_k(t) + \beta(\log p_i(t) - \log p_j(t)). \end{aligned}$$

We can draw the curves of the potential energy $\Psi(t) = (\Psi_i(t))_{i=1}^{n}$ of all cell states over time to illustrate the cell state-transition potential energy landscape from a dynamic point of view.

Based on Eq (2) which is the gradient flow of free energy [20, 21], we define the probability flow from vertex $j$ into vertex $i$ through edge $\{i, j\}$ between time stages $[t_l, t_{l+1}]$ as

$$\text{Flow}_{ij}(t_l, t_{l+1}) \equiv \int_{t_l}^{t_{l+1}} (F_j(p(t)) - F_i(p(t)))g_{ij}(p(t))dt, \quad (10)$$

which measures the total probability mass transporting from vertex $j$ into vertex $i$ through edge $\{i, j\}$ between adjacent time stages $[t_l, t_{l+1}]$: a positive value indicates a probability mass gain of vertex $i$ resulted from the flow of cells transiting from state $j$ into state $i$, while a negative value indicates a probability mass loss of vertex $i$ resulted from the flow of cells transiting from state $i$ into state $j$. If total probability mass is conservative (e.g., no cell proliferation is considered), we have

$$\text{Flow}_{ij}(t_l, t_{l+1}) = -\text{Flow}_{ji}(t_l, t_{l+1}).$$

The intuition of the probability flow definition is that, when potential energy difference $\Delta_{ji}(t) = F_j(p(t)) - F_i(p(t)) > 0$, cells tend to transit from a higher potential energy state $j$ to a lower potential energy state $i$, resulting in a probability mass gain of vertex $i$.

## Results

GraphFP models and infers cell differentiation as a cell state-transition process described by the nonlinear FPE on cell state-transition graph in continuous time (see Fig 1 and Methods for details).

In this study, we evaluated the performance of GraphFP using the time series scRNA-seq dataset of embryonic murine cerebral cortex development [26]. This dataset was analyzed by Tempora [12]. We downloaded the processed data, including the cell type annotations provided by Tempora [12]. The time series transcriptomic profile contains 6,316 cells collected at embryonic day 11.5 (E11.5), E13.5, E15.5 and E17.5. Overall, these cells represent neuronal development states from the early precursors (apical precursors (APs) and radial precursors (RPs)) to intermediate progenitors (IPs) and differentiated cortical neurons. Fig 2a illustrates the gold standard trajectory of the 4 major cell states curated by Tempora. Tempora identified 7 cell types by clustering and annotation methods: two AP/RP clusters denoted as "3-APs/RPs" and "5-APs/RPs", two IP clusters denoted as "4-IPs" and "7-IPs", two young neuron clusters denoted as "2-Young Neurons" and "6-Young Neurons", and one neuron cluster denoted as "1-Neurons" (Fig 2b).

## GraphFP reconstructs the cell state-transition energy landscape

We applied GraphFP to the embryonic murine cerebral cortex development scRNA-seq dataset based on the cell state/type labels of 7 clusters provided by Tempora. We estimated the cell state frequencies of the 7 cell states for each of the 4 time points separately. GraphFP first estimated parameters $\theta = \{\Phi, W\}$ of the free energy based on the adjoint method (Fig 2c and 2d).

In general, the static landscape of the estimated linear potential energies $\Phi$s shows a consistent understanding of the differentiation potencies of the 7 cell states. The early precursors states of "5-APs/RPs" and "3-APs/RPs" that mostly comprise cells at E11.5 have the highest two $\Phi$s of 0.059 and 0.054, respectively. The two IP clusters, "7-IPs" and "4-IPs", and one young neuron cluster, "6-Young Neurons", have the three intermediate-valued $\Phi$s of 0.052, -0.051, and 0.035, respectively. The differentiated cortical neuron clusters "1-Neurons" and "2-Young Neurons" have the lowest two $\Phi$s of -0.073 and -0.075, respectively (Fig 2c and 2e).

We modelled the cell state-transition energy landscape from a dynamical geometric point of view. The dynamic potential energy $\Psi(t)$ consists of not only the static linear part of $\Phi$, but also an interaction energy part as shown in Eq (8). It provides a global and holistic view of cell development process. For example, when only looking at the static landscape of $\Phi$, we found that the linear potential energy of "1-Neurons" ($\Phi_1 = -0.073$) is slightly higher than that of "2-Young Neurons" ($\Phi_2 = -0.075$). This is in conflict with our understanding that "1-Neurons" should have the lowest potential since it is located at the terminal node of the cell lineage (Fig 2a). However, when looking at the dynamic potential energy $\Psi(t)$, we found that "1-Neurons" has a strong inhibitory interaction over "2-Young Neurons" ($w_{21} = 0.26$ versus $w_{12} = 0.04$) which can upgrade the potential energy of "2-Young Neurons" during the process. The resultant dynamic potential energy of "2-Young Neurons" ($\Psi_2(t)$) surpasses that of "1-Neurons" ($\Psi_1(t)$) with higher value after time point E13 (Fig 2g and Left Panel of Fig 2h). The potential energy difference ($\Delta_{21}$) between "2-Young Neurons" and "1-Neurons" diverges with enlarging gaps as time evolves, especially in the latter time stages after time point E15.5 (Left Panel of Fig 2h). This is well consistent with our understanding that: (1) "2-Young Neurons" tends to transit to "1-Neurons" during cell development (Fig 2a), and (2) the transition from "2-Young Neurons" to "1-Neurons" mainly occurs at the late neurogenesis between E15.5 and E17.5 [26].

Further results on the linear potential energy, the cell-cell interactions and the dynamic potential energy will be provided in the following two sections.

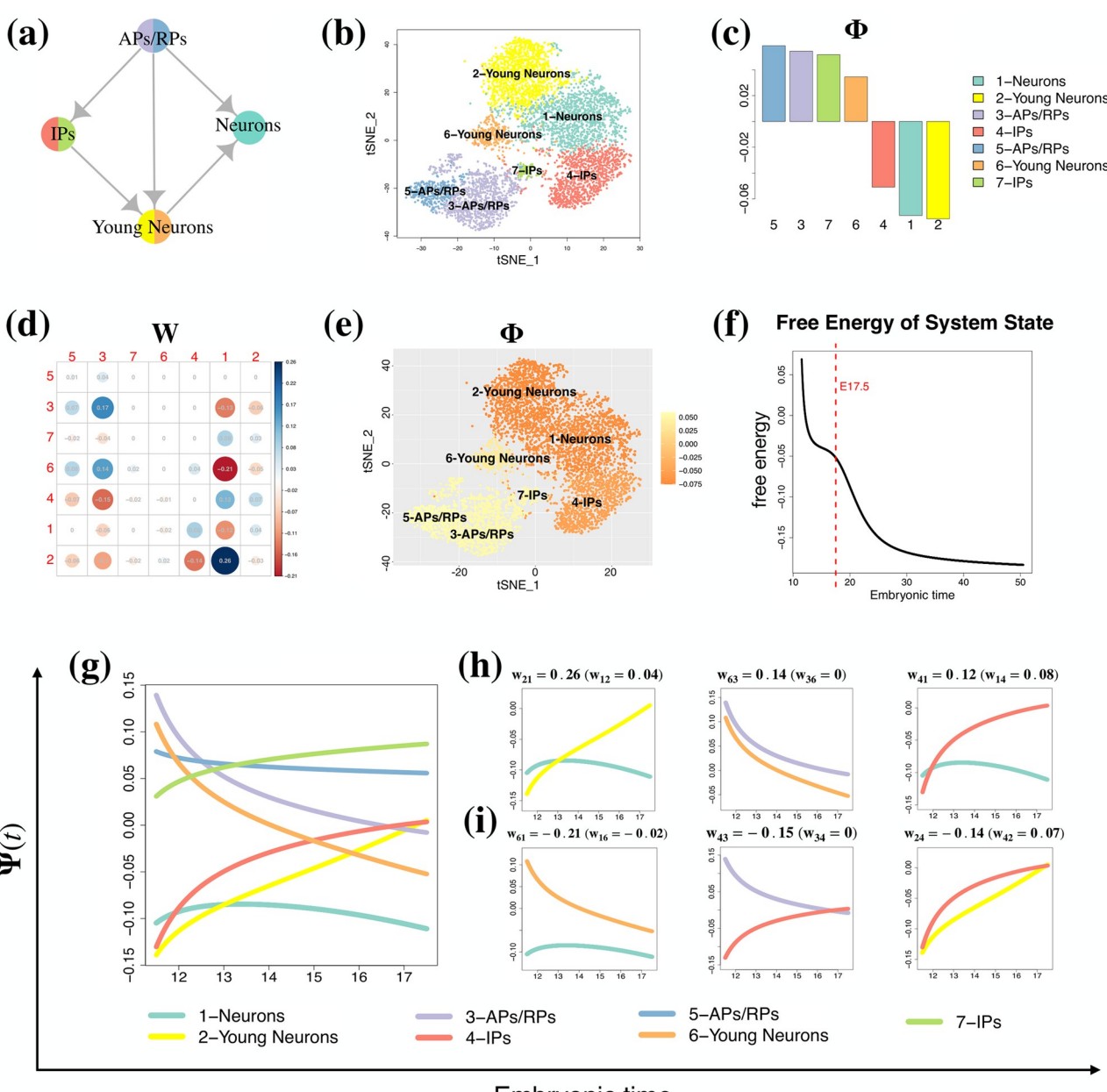

**Fig 2. GraphFP accurately reconstructs the cell state-transition energy landscape of the murine cerebral cortex dataset. (a)** The gold standard trajectory of embryonic murine cerebral cortex development. (b) The t-SNE plot of cells from the murine cerebral cortex dataset, colored by their cell-type labels. **(c)** GraphFP estimated the linear potential energy $\Phi$. **(d)** GraphFP estimated the cell-cell interaction matrix *W*. **(e)** Static linear potential energy landscape of cells on the t-SNE plot: cells are color-coded according to the linear potential energies $\Phi$s of their corresponding cell types. **(f)** The free energy (Eq (1)) of the system decreased over time. **(g)** The reconstructed potential energy landscape $\Psi(t)$ of cell types (colored curves) over time. **(h)** The potential energies of the cell state pairs with the top 3 highest positive values of cell-cell interaction strengths $w_{ij}$s: "2-Young Neurons ← 1-Neurons" (left panel), "6-Young Neurons ← 3-APs/RPs" (middle panel), and "4-IPs ← 1-Neurons" (right panel). **(i)** The potential energies of the cell state pairs with the top 3 lowest negative values of cell-cell interaction strengths $w_{ij}$s: "6-Young Neurons ← 1-Neurons" (left panel), "4-IPs ← 3-APs/RPs" (middle panel), and "2-Young Neurons ← 4-IPs" (right panel).

## The linear potential energy Φ by GraphFP quantifies cell differentiation potency

Computational quantification of cell differentiation potency (also known as cell stemness) is a challenging issue [18]. The pioneer work by Shi *et al.* [5] established a rigorous mathematical theory on quantifying cell stemness from scRNA-seq data based on continuous birth-death process.

Here, we demonstrated that the linear potential Φ estimated by GraphFP can be used to quantify the cell differentiation potency. In this study, each cell will be assigned the same linear potential value as that of its corresponding cell type/state. Shi *et al.* [5] also quantified the cell differentiation potency at the cluster level (e.g., cell state and cell type), which makes the results more accurate and robust. Quantifying the cell differentiation potency at single-cell level is still difficult, as the single-cell gene expression profiles are known to be error-prone due to various technique issues [31].

Following the study in Shi *et al.* [5], we tested whether our linear potential energies for pluripotent stem cells (at early time point) are higher than those for differentiated cells (at latter time point). It is clearly shown that, cells from samples collected at earlier time stages tend to have higher potential Φ and vice versa (Fig 3a). When using the one-sided Wilcoxon ranksum statistic as applied by Shi *et al.* [5], we confirmed with highly statistically significant results that the linear potential values of cells sampled at the earliest time stage E11.5 are higher than those cells sampled at the subsequent time stages E13.5 ($p < 1.554e − 07$), E15.5 ($p < 2.2e − 16$), and E17.5 ($p < 2.2e − 16$), respectively.

Next, we checked the linear potential energies of the cell types with their pseudo-time during cell development process. Tempora [12] provided each cell type with a temporal score by adjusting its cell composition from each time point such that a cell type containing more cells from an early time point will have a lower score and vice versa. We therefore used the temporal scores as the pseudo-time for the 7 cell types. It is clearly demonstrated that the addictive inverse values of linear potential energy are strongly correlated with the temporal scores (Fig 3b, Pearson correlation coefficient = 0.91), further confirming that the linear potential energy well quantifies cell stemness.

It is worth noting that the linear potential energy Φ (Fig 3c) is different from the stationary distribution $p_{ss}$ of cell types (Fig 3d). The stationary distribution $p_{ss}$, which is the cell type frequencies or cell densities calculated using the merged data across all time points, is often used to construct the stationary energy landscape $U_{ss} \equiv -\log p_{ss}$ in scRNA-seq data analysis [6]. However, as pointed out by Shi *et al.* [5], the stationary energy landscape $U_{ss}$ is the equilibrium potential induced by diffusion without birth and death. In some extent, the linear potential energy Φ by GraphFP is an analogy of the cell potential $V(x)$ proposed by Shi *et al.* [5], which was taken as their quantification of cell differentiation potency.

## GraphFP delineates cell-cell interactions

We quantified the cell-cell interactions and intercellular communication during embryonic murine cerebral cortex development using the cell-cell interaction matrix $W$ estimated by GraphFP. The $W$ measures the cell-cell interaction strength between each pair of two cell types/states, one as the signalling sender and the other as the signalling receiver (see Eq (9) and section **"Reconstruction of cell developmental energy landscape and modelling of cell-cell interactions"** in **Methods** for its biological interpretations).

The estimated $W$ is a sparse matrix with most elements having values equal or close to zero (Fig 2d), indicating a majority of the pairs having no or weak interactions. For example, the cell types "5-APs/RPs" and "7-IPs" have no mutual interactions with $w_{57} = 0$ and $w_{75} = −0.02$

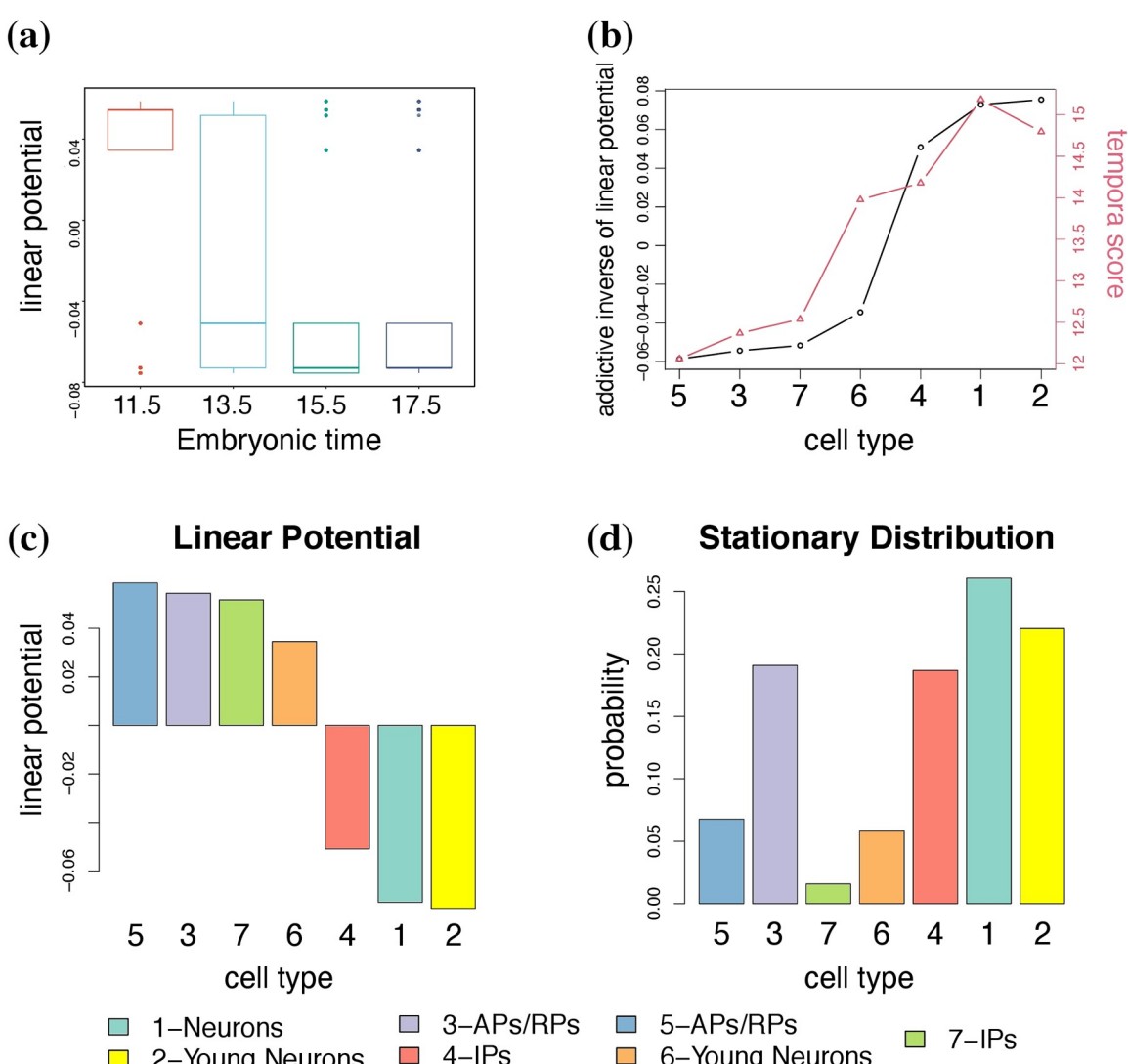

**Fig 3. The linear potential energy Φ quantifies cell differentiation potency. (a)** Boxplot of the linear potential energies of cells sampled different time stages of the embryonic murine cerebral cortex development. **(b)** Trend in the addictive inverse of linear potential (circular points connected by black lines with y axis on left-hand side) and temporal score (triangle points connected by red lines with y axis on right-hand side) across cell types. **(c)** The linear potential energy Φ estimated by GraphFP. **(d)** The stationary probability distribution $p_{ss}$ of the cell types.

(Fig 2d). Furthermore, the estimated cell-cell interaction strengths in the first row of **W** that corresponds to "5-APs/RPs" are zero or close to zero with $|w_{ij}| \leq 0.04$, making the contributions from its interaction term in Eq (8) negligible. As such, the dynamic potential energy $\Psi_5(t)$ is dominated by its linear potential energy ($\Phi_5$) with a resultant flattening potential energy curve (Fig 2g).

We also observed a number of strong cell-cell interactions with large $w_{ij}$s deviating from zero. Cell states other than "5-APs/RPs" and "7-IPs" have at least one $w_{ij}$ with strong interaction strength (e.g., $|w_{ij}| > 0.1$), resulting in the deviation of their potential energies $\Psi(t)$s from their linear potential energies $\Phi$s largely driven by their interaction energies with sharpened potential energy curves (Fig 2g).

We further examined cell state pairs with strong interactions.The pairs of "2-Young Neurons ← 1-Neurons" ($w_{21}$ = 0.26), "6-Young Neurons ← 3-APs/RPs" ($w_{63}$ = 0.14) and "4-IPs ← 1-Neurons" ($w_{41}$ = 0.12) have the top 3 highest positive values of $w_{ij}$s (Fig 2d), indicating that the sender cell types ("1-Neurons", "3-APs/RPs" and "1-Neurons") pass strong inhibitory signalling to their corresponding receiver cell types ("2-Young Neurons", "6-Young Neurons" and "4-IPs", respectively). Their potential energy differences ($|\Delta_{ij}|$s) diverge with enlarging gaps as time evolves (Fig 2h), resulting in that cells tend to transit in one direction from the cell state with higher potential energy to the cell state with lower potential energy, only rarely transiting in the reverse direction.In particular, "2-Young Neurons" tends to transit to "1-Neurons", "3-APs/RPs" tends to transit to "6-Young Neurons", "4-IPs" tends to transit to "1-Neurons". These results are consistent with our understanding of the cell development process depicted in Fig 2a.

On the other hand, the pairs of "6-Young Neurons ← 1-Neurons" ($w_{61}$ = −0.21), "4-IPs ← 3-APs/RPs" ($w_{43}$ = −0.15) and "2-Young Neurons ← 4-IPs" ($w_{24}$ = −0.14) have the top 3 lowest negative values of $w_{ij}$s, indicating that the sender cell types ("1-Neurons", "3-APs/RPs" and "4-IPs") pass strong stimulatory signalling to their receiver cell types ("6-Young Neurons", "4-IPs" and "2-Young Neurons", respectively).In particular, the potential energy differences ($|\Delta_{ij}|$s) of the pair "6-Young Neurons: 1-Neurons" and the pair "3-APs/RPs: 4-IPs" converge with shrinking gaps as time evolves (Fig 2i), making the transitions between the paired cell states approaching to equilibrium in both directions. This result is consistent with our probability flow (Fig 4), where the net probability flows from "6-Young Neurons" to "1-Neurons" as well as from "3-APs/RPs" to "4-IPs" gradually decrease over time. The potential energy difference between "4-IPs" and "2-Young Neurons" starts from a small value close to zero at time point E11.5, then gradually increases to its largest gap at E14, and then gradually declines to zero again at E17.5. This result indicates that the transition from "4-IPs" to "2-Young Neurons" mainly occurs at the intermediate time region from E13.5 to E15.5, which is consistent with our understanding that the "4-IPs" cells are the intermediate progenitors of cell development (Fig 2a).

Based on our calculation using GraphFP, we also confirmed that free energy (Eq (1)) of the system decreased over time (Fig 2f), which is consistent with accepted mathematical theory [20, 21]. However, according to our calculation, free energy of the system did not converge to its minimum free energy state at time point E17.5 when the experiment ended (see the vertical dashed red line in Fig 2f). We predicted from Fig 2f that the system would reach its minimum free energy state after time point E30.

## GraphFP faithfully charts the probability flows of cell state-transitions during cell development

We next examined the ability of GraphFP to quantify the dynamics of cell state-transitions during embryonic murine cortical development by calculating the probability flows (Eq (10)) between each time intervals of the adjacent time stages (Fig 4). In the early stage from E11.5 to E13.5, cells mainly transit from the early precursors of "3-APs/RPs" and "5-APs/RPs" to the intermediate progenitor "4-IPs" and the two neuron clusters of "2-Young Neurons" and "1-Neurons". In the middle stage from E13.5 to E15.5, the intermediate progenitor "4-IPs" joins in with "3-APs/RPs" and "5-APs/RPs" as the major source clusters that transit to two neuron clusters, "2-Young Neurons" and "1-Neurons". Meanwhile, as a source cluster, "2-Young Neurons" starts to transit to "1-Neurons", and in the latter stage from E15.5 to E17.5, "4-IPs" takes a leading role in transiting to the neuron cluster of "1-Neurons" and young neuron clusters, "2-Young Neurons" and "6-Young Neurons". Meanwhile, "2-Young

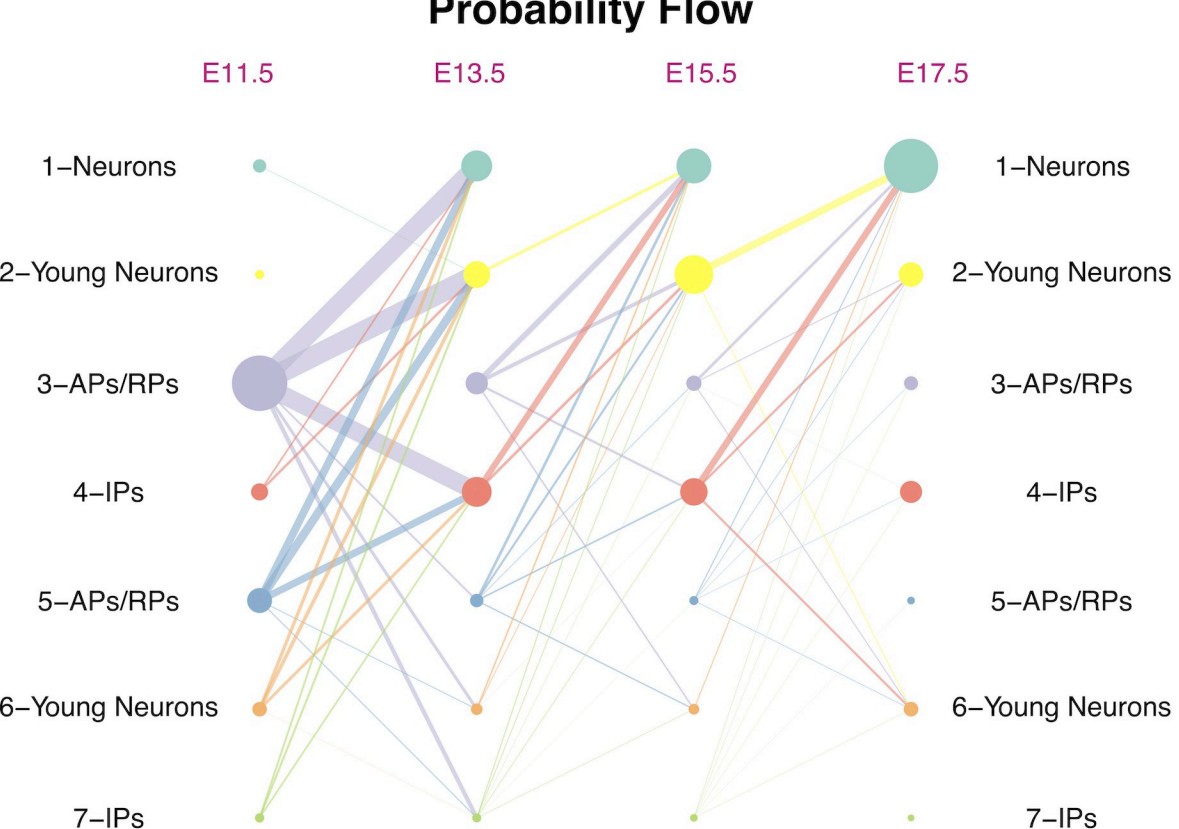

**Fig 4. GraphFP charts the probability flows of cell state-transitions.** The circle point represents cell type (point size is proportional to the cell-type frequency at each time point); the line between cell types represents probability flow from source cell type to target cell type (line width is proportional to the value of probability flow).

Neurons" continues as one of the major source clusters transiting to "1-Neurons" (Fig 4). Compared with the gold standard trajectory shown in Fig 2a by Tempora [12], we identified a new path whereby the IP cells of cluster "4-IPs" transit to neuron cells of cluster "1-Neurons", as confirmed by Yuzwa *et al.* [26], who reported that cortical RPs divide asymmetrically from E11.5 to E17.5 to generate neurons directly or indirectly via transit-amplifying cells of IPs.

### Cell-cell interactions drive the stochastic and nonlinear dynamics of cell development

GraphFP explicitly models cell-cell interactions with a nonlinear quadratic interaction term in the free energy (Eq (1)). To account for cell-cell interactions, we evaluated GraphFP on its ability to fit the experimental data (Fig 5a) and recover held-out time points (Fig 5b–5d) with cell-cell interaction term ($W \neq 0$; solid lines in Fig 5) and without cell-cell interaction term ($W = 0$; dashed lines in Fig 5). To evaluate the performance on estimation accuracy, we applied Kullback-Leibler divergence (KLD) to measure the difference between the estimated probability distribution with/without interaction term and true probability distribution at each time points (Table 1). A lower KLD value is indicative of better performance.

First, we applied GraphFP to the embryonic murine cerebral cortex development scRNA-seq dataset using all 4 time points. Based on the estimated parameters $\theta^*$, we calculated the

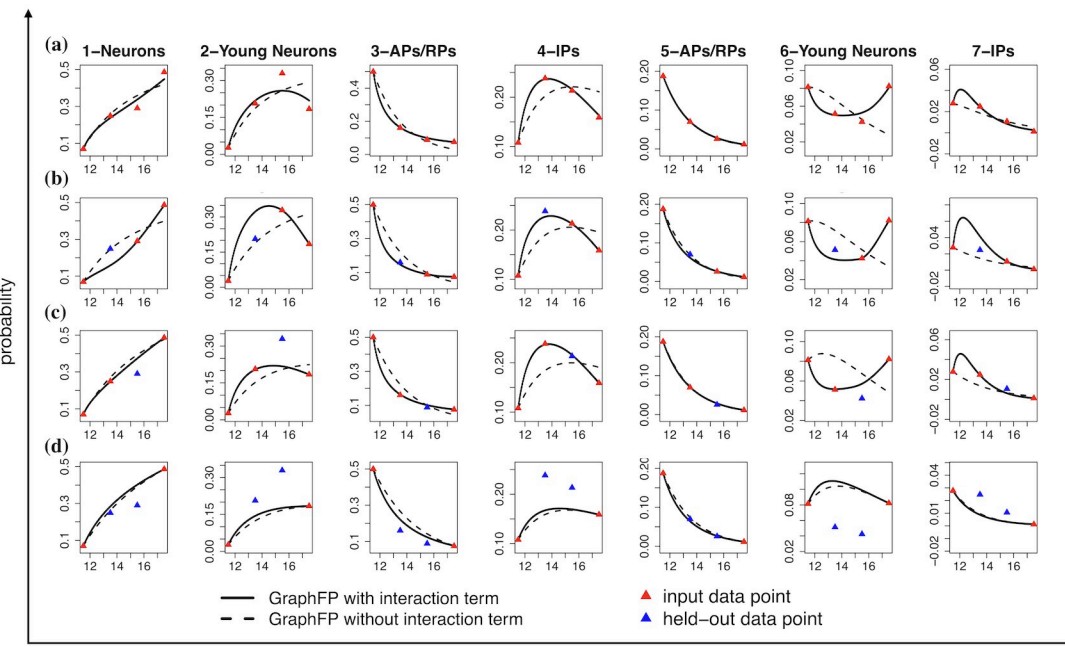

**Fig 5. GraphFP accurately quantifies the stochastic dynamics of the cell type frequencies by modelling cell-cell interactions.**
GraphFP calculated the stochastic dynamics of the cell type frequencies $p(t)$ with cell-cell interaction term ($W \neq 0$; solid lines) and without cell-cell interaction term ($W = 0$; dashed lines). Triangle points are the estimated cell type frequencies at each time point where red represents the input data point to GraphFP, while blue represents the held-out data point to GraphFP. **(a)** Using all 4 time points as input. **(b)** Held-out E13.5. **(c)** Held-out E15.5. **(d)** Held-out E13.5 and E15.5.

stochastic dynamics of the cell type frequencies $p(t)$ on probability simplex in continuous time $t(> t_1)$ according to Eq (2) with given initial point of $\hat{p}(t_1) = p^1$. Overall, GraphFP with cell-cell interaction term outperforms GraphFP without cell-cell interaction term on the fitting of the nonlinear curves for the 7 clusters (Table 1), especially for "2-Young Neurons", "4-IPs" and "6-Young Neurons" (Fig 5a).

Next, we applied GraphFP to the scRNA-seq datasets of (i) one held-out time point E13.5 (Fig 5b) and (ii) one held-out time point E15.5 (Fig 5c), separately. GraphFP with cell-cell interaction term always outperforms GraphFP without cell-cell interaction term on both non-linear curve fitting and held-out time point recovering except for one comparison on Held out E13.5 dataset at time stage E13.5. (Table 1).

**Table 1. Evaluation of GraphFP's performance on quantifying the stochastic dynamics of cell-type frequencies with cell-cell interaction term ($W \neq 0$) and without cell-cell interaction term ($W = 0$) on the murine cerebral cortex dataset.**

| KLD | With all time points | | Held out E13.5 | | Held out E15.5 | | Held out E13.5 and E15.5 | |
|---|---|---|---|---|---|---|---|---|
| | with | without | with | without | with | without | with | without |
| E13.5 | **0.0007** | 0.0101 | 0.0340 | **0.0217** | **7.0020e-05** | 0.0377 | **0.0675** | 0.0900 |
| E15.5 | **0.0133** | 0.0168 | **0.0009** | 0.0119 | **0.0372** | 0.0518 | **0.1036** | 0.1130 |
| E17.5 | **0.0056** | 0.0861 | **0.0036** | 0.0785 | **5.0603e-05** | 0.0240 | **1.9569e-06** | 2.0707e-06 |

The Kullback-Leibler divergence (KLD) distance was used to measure the difference between the estimated probability distribution by GraphFP and true probability distribution at each time points (E13.5, E15.5, E17.5).

Finally, we applied GraphFP to the scRNA-seq data set of two held-out time points, E13.5 and E15.5 (Fig 5d). It is not surprising that both models drop their accuracies markedly on recovering the held-out time points.In addition, the results by GraphFP with cell-cell interaction term still outperform those by GraphFP without cell-cell interaction term (Table 1 and Fig 5d).

As shown in Fig 5, our results illustrate that the stochastic and nonlinear dynamics of cell development are not merely determined by the linear potential energies Φs, but also driven by nonlinear cell-cell interactions. Specifically, the evolving probability frequencies of cell types can be nonmonotonic (e.g., "2-Young Neurons", "4-IPs", "6-Young Neurons" in Fig 5).Meanwhile, time series scRNA-seq data with cells profiled at more time points will provide more temporal information to recover the biologically complex dynamic processes.

## GraphFP is robust to input data

As also shown in Fig 5a–5c, our results illustrate that GraphFP robustly recovers the stochastic and nonlinear dynamics of cell development by using all datasets or datasets with one held-out time point.

Since GraphFP works on cells with cluster labels or cell type annotations, we then examined whether GraphFP is sufficiently robust to account for the uncertainty presented in the clustering or annotation methods. In the above sections, we have illustrated the outputs of GraphFP based on the labelling of 7 clusters with a fine resolution provided by Tempora [12]. Here, we further grouped the cells into 4 clusters with a coarse resolution as follows: "A-Neurons" constituted by cells from "1-Neurons"; "B-Young Neurons" constituted by cells from "2-Young Neurons" and "6-Young Neurons"; "C-APs/RPs" constituted by cells from "3-APs/RPs" and "5-APs/RPs"; and "D-IPs" constituted by cells from "4-IPs" and "7-IPs". We compared the results of GraphFP based on the labelling of 7 cell types and the labelling of 4 cell types (Fig 6). To make the results comparable, we aggregated the results based on the labelling of 7 cell types by averaging the results from i) "3-APs/RPs" and "5-APs/RPs", ii) "2-Young Neurons" and "6-Young Neurons" and iii) "4-IPs" and "7-IPs", separately, resulting in the same dimensions as those based on the labelling of 4 cell types. It should be noted that the results based on 4 cell types and the aggregated results based on 7 cell types are consistent with similar patterns of linear energies Φs (Fig 6a and 6d), interaction matrices $W$s (Fig 6b and 6e), and probability flows (Fig 6c and 6f). In addition, GraphFP is robust to the hyper-parameter choices in wide ranges.

## The computational cost of GraphFP

We examined the impact of the number of cell types ($n$) on the computational cost of GraphFP. When working on the murine cerebral cortex dataset with 7 cell types ($n = 7$) and 4 time stages, the runtime of GraphFP was around 3 minutes for each task on a personal laptop (MacBook Pro with CUP 2.4 GHz Intel Core i5 and Memory 8 GB 2133 MHz LPDDR3) (S1 Table). In our implementation, we set $\lambda_l = 1000 (l \in 2, 3, 4)$, $\beta = 0.001$, the learning rate as $\alpha = 0.01/\lambda_l$ and $Integral\_step$ as 0.1.

We next examined GraphFP on another time series scRNA-seq dataset of the mouse spinal cord injury healing process provided by [32] (see the detailed results in S2 Text). The new dataset contains 13 clusters (cell types) and 4 time points. We applied GraphFP to this dataset on the same computer with the same hyper-parameter settings as those for the murine cerebral cortex dataset. GraphFP still achieved accurate and robust reconstruction of cell state-transition energy landscape on this dataset (S2 Text), and also achieved a reasonable performance on computational speed: the runtime for each task was around 9 minutes (S2 Table).

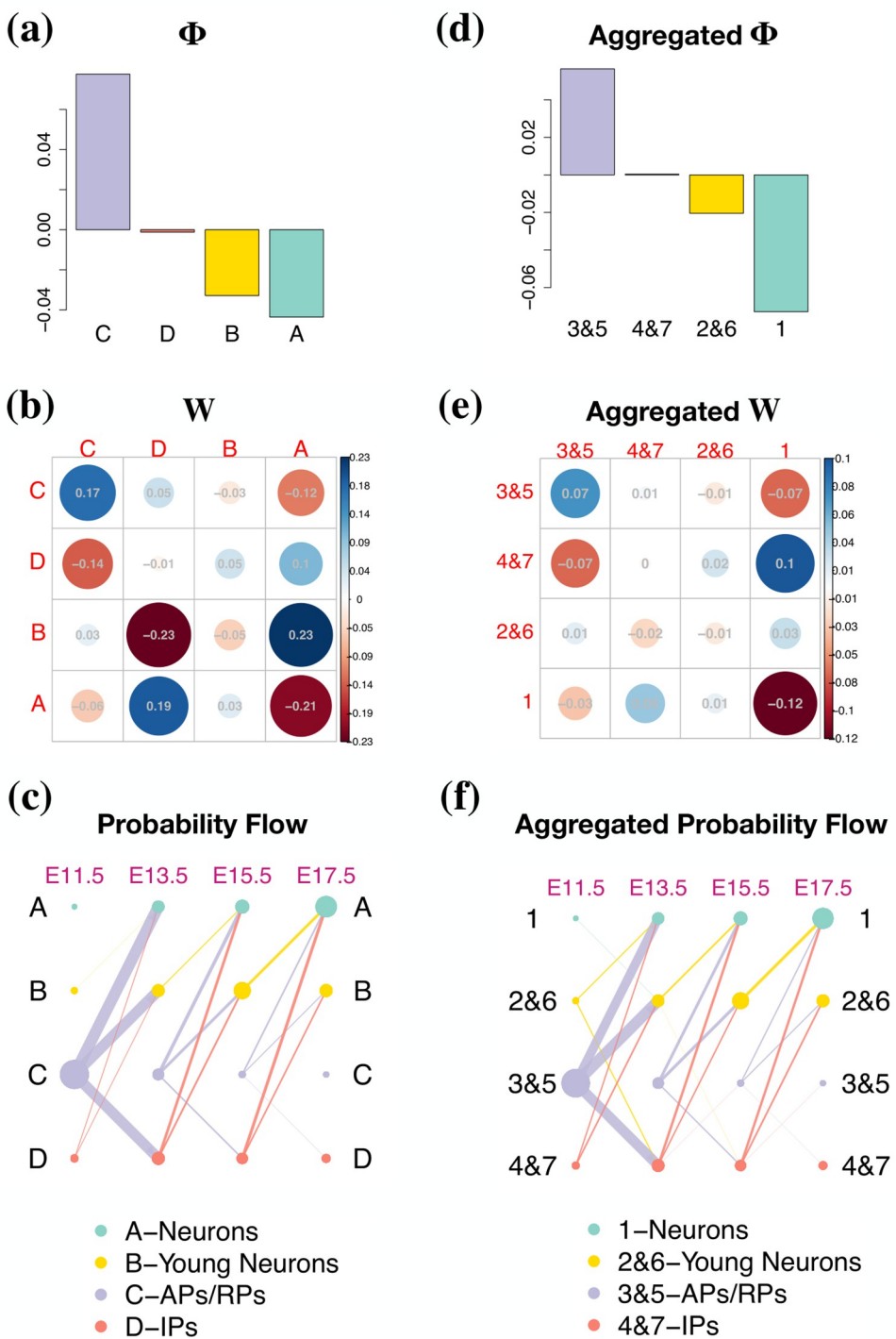

**Fig 6. GraphFP is robust to uncertainty presented in cell type labels.** GraphFP was applied to the murine cerebral cortex dataset based on the labelling of 4 cell types with a coarse resolution (**a-c**) and the labelling of 7 cell types with a fine resolution (**d-f**), separately. The estimated **Φ** (**a**), **W** (**b**) and charted probability flow (**c**) by GraphFP based on the labelling of 4 clusters ("A-Neurons", "B-Young Neuron", "C-APs/RPs", "D-IPs"). Aggregated results of the estimated **Φ** (**d**), **W** (**e**) and charted probability flow (**f**) by GraphFP based on the labelling of 7 clusters, averaging the results from i) "3-APs/RPs" and "5-APs/RPs", ii) "2-Young Neurons" and "6-Young Neurons" and iii) "4-IPs" and "7-IPs", separately, resulting in the same dimensions as those based on the labelling of 4 cell types.

Based on the above experiments, we found that the computational speed of GraphFP is sustainable for tasks with moderate number of cell types. Meanwhile, the computation of GraphFP might be problematic for large $n$ (e.g., $n > 100$) at current settings. As we applied the complete cell state-transition graph, the degree of freedom (e.g., the parameters of the cell-cell interaction matrix $W$) will grow in the order of $O(n^2)$, making the computation difficult. However, this problem is solvable. One way to solve this problem is to take advantage of the sparse structure of the cell-cell interaction matrix $W$. As we have already noted, the estimated $W$s of both the murine cerebral cortex dataset (Fig 2d) and the mouse spinal cord injury dataset (Fig A(b) in S2 Text) are sparse. Therefore, we can solve this problem by adding a L1 regularization term of the matrix $W$ to the loss function to enforce $W$ to be sparse. We plan to pursue this topic in our future work.

On the other hand, in practice, for large number of cell types, we can trade off the estimation accuracy and the information of cell-cell interactions for runtime performance. GraphFP without the cell-cell interaction term will be efficient for large number of cell types since the degree of freedom (e.g., the parameters of linear potential energy $\Phi$) will grow in the order of $O(n)$. For example, the runtimes of GraphFP without cell-cell interaction term ($W = 0$) for both the murine cerebral cortex dataset (S1 Table) and the mouse spinal cord injury dataset (S2 Table) are all less than 20 seconds.

## Discussion

Modelling of cell development has long been a key goal of systems biology. The Waddington landscape is a classic metaphor for describing cell development. Mathematical framework of cell developmental energy landscape has been developed to study the dynamics of cell state-transitions from gene regulatory network (GRN) based perspective (e.g., [33–36]) and state manifold based perspective (e.g., [5, 37, 38]) (see [18] for a recent review of the two approaches). Traditional GRN-based landscape can be hindered by the computational issue raised by high-dimensional GRNs. Recently, a model-based dimension reduction approach of the landscape (DRL) was proposed to construct a low-dimensional energy landscape of high-dimensional GRNs [36], which overcomes the limitations of traditional methods. Although great success has been achieved, the GRN-based landscape depends on prior biological knowledge of the underlying GRN. When the information of GRNs is unavailable or not complete, the state manifold based landscape will be constructed, especially for scRNA-seq data analysis. The state manifold based methods model the cell development with stochastic Markov process and/or drift-diffusion PDE, where cell states (e.g., cell types and cell clusters) represent the local attractors of the underlying dynamic systems [18].

In this study, we propose GraphFP, a state manifold based computational framework, to reconstruct the complex potential energy landscape and infer the stochastic dynamics of cell state-transition during cell development. GraphFP models cell development based on the diffusion process in a discrete spectrum of states [19–21]. It can be viewed from the lens of dynamic optimal transport on networks as solving an optimal control problem to minimize the kinetic energy of flow between adjacent time points [14, 39]. The FPE of GraphFP can be characterized as a gradient flow of free energy when the probability simplex of discrete states is equipped with the discrete $L_2$-Wasserstein metric defined on the graphs [19–21]. Beyond its clear theoretical importance, GraphFP has enabled critical insight into nonlinear dynamic cell state-transition, as well as cell-cell interactions during cell development. We demonstrated that the cell-cell interaction part of GraphFP plays a key role in capturing the stochastic dynamic of the cell-type frequencies on both the murine cerebral cortex dataset (Table 1 and Fig 5) and the mouse spinal cord injury dataset (S2 Text).

GraphFP has the following strengths over existing methods. First, GraphFP models the dynamics of cell clusters (e.g., cell states and cell types) on a discrete state space. In contrast, methods, such as Waddington-OT [15], TrajectoryNet [16] and PRESCIENT [17], modelled the dynamics of individual cells with drift-diffusion equations on a continuous state space. With the dramatic increase in amount and size of scRNA-seq data, the cluster-based approaches, which work on a relatively small number of clusters that usually represent annotated cell types, warrant both scalability to large-scale scRNA-seq data and ease of biological interpretability [12].

Second, GraphFP is built on a nonlinear model that explicitly takes into account cell-cell interactions in free energy. The current computational methods for inferring cell-cell interactions from single-cell data are mainly based on machine learning or statistics, relying heavily on the domain knowledge as learning materials [22, 23]. On the other hand, GraphFP provides an alternative and model-based approach to decipher cell-cell interactions that drive cell development. In contrast, the underlying models of both Waddington-OT [15] and PRESCIENT [17] are only able to characterize cell state-transition on the static potential energy landscape driven by random noises, failing to account for cell-cell interactions. Although able to reconstruct nonlinear development landscape, TrajectoryNet was based on the neural network framework without explicit system models, thus lacking biological interpretability.

Nonetheless, some aspects still need to be improved. Firstly, the current GraphFP does not account for cell proliferation during cell development, which may result in that probability masses are not conservative over time. We can solve this problem by adopting the unbalanced optimal transport framework that has been used by Waddington-OT [15] and TrajectoryNet [16] to quantify cell proliferation. Secondly, as the existing time series scRNA-seq methods such as Waddington-OT [15] and Tempora [12], the current GraphFP works in an off-line fashion such that the cell clustering and annotation are performed on the entire data by merging cells from all time points together. This approach offers an unbiased, comprehensive and quantitative definition of discrete cell types. However, with the emerging large-scale scRNA-seq data, it may be computationally cumbersome to cluster the massive and continually arriving scRNA-seq datasets as a whole. Therefore, developing an on-line framework of GraphFP that can cluster and annotate the single-cell time series scRNA-seq data in different batches in a serial fashion should be an interesting topic. The newly developed single-cell data analysis tools such as the on-line integration method online iNMF [40] and the cell type annotation method scArches based on transfer learning [41] can be adopted.

## Supporting information

**S1 Text. Details for the parameter estimation of GraphFP.** This document provides detailed description of the parameter estimation and pseudocode for the GrapFP algorithm.
(PDF)

**S2 Text. Application of GraphFP to the mouse spinal cord injury dataset.** Fig A. GraphFP reconstructs the cell state-transition energy landscape on the mouse spinal cord injury scRNA-seq dataset. Fig B. The linear potential energy $\Phi$ quantifies cell differentiation potency. Table A. Evaluation of GraphFP's performance on quantifying the stochastic dynamics of cell type frequencies with cell-cell interaction term ($W \neq 0$) and without cell-cell interaction term ($W = 0$) on the mouse spinal cord injury dataset.
(PDF)

**S1 Table. Runtimes of GraphFP on the murine cerebral cortex dataset.**
(DOCX)

**S2 Table. Runtimes of GraphFP on the mouse spinal cord injury dataset.**
(DOCX)

## Author Contributions

**Conceptualization:** Qi Jiang, Lin Wan.

**Data curation:** Qi Jiang.

**Formal analysis:** Qi Jiang, Lin Wan.

**Funding acquisition:** Shuo Zhang, Lin Wan.

**Investigation:** Qi Jiang, Shuo Zhang, Lin Wan.

**Methodology:** Qi Jiang, Shuo Zhang, Lin Wan.

**Software:** Qi Jiang.

**Supervision:** Shuo Zhang, Lin Wan.

**Validation:** Qi Jiang, Shuo Zhang, Lin Wan.

**Visualization:** Qi Jiang.

**Writing – original draft:** Qi Jiang, Lin Wan.

**Writing – review & editing:** Qi Jiang, Shuo Zhang, Lin Wan.

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
