## [Decision Letter · Decision Letter 0]

6 Sep 2021

Dear Dr. Wan,

Thank you very much for submitting your manuscript "Dynamic inference of cell developmental complex energy landscape from time series single-cell transcriptomic data" for consideration at PLOS Computational Biology.

As with all papers reviewed by the journal, your manuscript was reviewed by members of the editorial board and by several independent reviewers. In light of the reviews (below this email), we would like to invite the resubmission of a significantly-revised version that takes into account the reviewers' comments.

We cannot make any decision about publication until we have seen the revised manuscript and your response to the reviewers' comments. Your revised manuscript is also likely to be sent to reviewers for further evaluation.

Sincerely,

Qing Nie

Associate Editor

PLOS Computational Biology

Douglas Lauffenburger

Deputy Editor

PLOS Computational Biology

Reviewer's Responses to Questions

**Comments to the Authors:**

Reviewer #1: In this work, the authors propose an approach to infer cell state transition dynamics and uncover cell-cell interactions from time series single-cell RNA-seq data, by constructing the energy landscape of the system. Based on a nonlinear Fokker-Planck equation on Graph based model, they solve the model inference problem in a dynamic optimal transport framework. They further illustrate the validity of their approach by applying it to a time series scRNA-seq data set of embryonic murine cerebral cortex development.

Overall, this is an interesting work, aiming to address a critical problem in dynamic model inference and data-driven approach in biology. However, I have some concerns that need to be addressed.

1, The authors assume that the cell-cell interaction matrix W is symmetric, what does this assumption mean in biology? And why is this a reasonable assumption? For example, should cells with different differentiation potency have symmetric interactions?

2, Fig.2 (a)(b)(c), the use of color for different cell types is very confusing. Please try to make them consistent.

3, It’s interesting to get cell-cell interaction matrix W as in Fig.2. However, the biological indication needs to be further discussed. For example, from Fig. 2 (g and h) it is found that cell type 3 (Aps/RPs) has positive interaction with cell type 6 (Young Neurons), and negative interaction with cell type 4 (IPs) or 2 (Young Neurons). What could be the biological meaning here based on the linage commitment route in Fig. 2(a)? Why do the two Young Neurons populations (cell type 6 and 2) have opposite types of interactions with cell type 3 (Aps/RPs)?

4, The energy landscape here is defined for each cell type, could it be expanded to single cell? i.e., each cell has a potential energy?

5, Current work focus on the cell-cell interactions, and molecular expression data is used only for estimating probability of cell types at different time point. As we know, the molecular interactions are important to cell fate transition determinations, and corresponding landscape construction approaches have been proposed (Kang, Advanced Science, 8, 2003133 (2021)). The authors may want to discuss the relationship between the two types of approaches.

Reviewer #2: The authors consider the dynamic inference of an evolving cell population from the time series scRNA-seq data. The main idea of their approach is to apply the optimal transport theory on graph with a free energy consists of the potential, mean field interaction, and the entropy terms. The parameters describing the potential and interaction terms are estimated by fitting the model to the available data. The computation applied to embryonic murine cerebral cortex development gives reasonable results. The literature review is also fair. Overall, I think the paper is interesting and present simple yet useful tool to analyze the time series scRNA-seq data.

Some major and minor points are as below.

Major:

1. The computational cost and robustness. The authors state that the method will be applied to the cell states/types in a complete graph. If it is only for cell types, which have a small number, the DOF is small and the computation cost should be light. If it is for the whole cell states, the DOF might be huge and the computation is difficult to be done. BTW, in different time points, how will the authors choose the common cell states if they do not do grouping at first? How will the different grouping affect the result?

2. Quantification of the cell stemness. This is an important issue in scRNA-seq data analysis. Will the proposed method give a quantification on this problem? I would like to see some tests and comments on this point.

3. The authors choose the optimal transport (OT) approach and and Wasserstein distance to the scRNA-seq data analysis, which is a hot topic in this area and machine learning community. Yes, it is fancy to take OT. But it is also possible to utilize other simpler model and optimization approach to accomplish similar task. I wonder whether the authors have some reason to take OT instead of other methods. Some comments and discussion on this point should be added.

Minor:

1. Some typo or gramatical errors.

Line 73: double periods in the text.

Line 121: as follows

**Have the authors made all data and (if applicable) computational code underlying the findings in their manuscript fully available?**

Reviewer #1: Yes

Reviewer #2: Yes

PLOS authors have the option to publish the peer review history of their article (what does this mean?). If published, this will include your full peer review and any attached files.

Reviewer #1: No

Reviewer #2: No
---

## [Decision Letter · Decision Letter 1]

8 Dec 2021

Dear Dr. Wan,

Thank you very much for submitting your manuscript "Dynamic inference of cell developmental complex energy landscape from time series single-cell transcriptomic data" for consideration at PLOS Computational Biology. As with all papers reviewed by the journal, your manuscript was reviewed by members of the editorial board and by several independent reviewers. The reviewers appreciated the attention to an important topic. Based on the reviews, we are likely to accept this manuscript for publication, providing that you modify the manuscript according to the review recommendations.

Sincerely,

Qing Nie

Associate Editor

PLOS Computational Biology

Douglas Lauffenburger

Deputy Editor

PLOS Computational Biology

[LINK]

Reviewer's Responses to Questions

**Comments to the Authors:**

Reviewer #1: The authors have addressed my concerns thoroughly. I have only a few minor comments.

1, Line 483, is ref. 37 the right reference here? It seems that ref. 37 is not talking about data-driven landscape.

2, The final word of the paper (Line 533), “adopt” should be adopted?

3, Fig. 3 (c and d), the y axis seems to be labeled incorrectly.

Reviewer #2: The authors responded to my concerns perfectly. Now I recommend it for publication.

**Have the authors made all data and (if applicable) computational code underlying the findings in their manuscript fully available?**

Reviewer #1: Yes

Reviewer #2: Yes

PLOS authors have the option to publish the peer review history of their article (what does this mean?). If published, this will include your full peer review and any attached files.

Reviewer #1: No

Reviewer #2: No

Figure Files:

Data Requirements:

Reproducibility:

References:

---

## [Editor Report · Decision Letter 2]

10 Jan 2022

Dear Dr. Wan,

We are pleased to inform you that your manuscript 'Dynamic inference of cell developmental complex energy landscape from time series single-cell transcriptomic data' has been provisionally accepted for publication in PLOS Computational Biology.

Best regards,

Qing Nie

Associate Editor

PLOS Computational Biology

Douglas Lauffenburger

Deputy Editor

PLOS Computational Biology

Feilim Mac Gabhann

Editor-in-Chief

PLOS Computational Biology

---

## [Editor Report · Acceptance letter]

20 Jan 2022

PCOMPBIOL-D-21-00954R2 

Dynamic inference of cell developmental complex energy landscape from time series single-cell transcriptomic data

Dear Dr Wan,

I am pleased to inform you that your manuscript has been formally accepted for publication in PLOS Computational Biology. Your manuscript is now with our production department and you will be notified of the publication date in due course.

With kind regards,

Agnes Pap
